# Uric acid levels mediate the association between four dietary indices and kidney stones in US adults: A cross-sectional study of NHANES 2007–2018

**Jinlong Cao**[ID], **Tianyuan Zhai, Lingyu Guo, Yue Chong, Qi Chen, Qian Wang, Delai Fu, Li Xue\*, Feng Li\***

Department of Urology, The Second Affiliated Hospital of Xi'an Jiaotong University, Xi'an, China,

\* xueli1979@xjtu.edu.cn (L) lifeng.uro@xjtu.edu.cn (FL)

## Abstract

### Background

Diet and uric acid are closely linked to the formation of kidney stones. However, the specific dietary indices associated with kidney stone risk and the mediating role of uric acid remain unclear. This study investigates the association between four dietary indices and kidney stone risk while exploring the mediating role of uric acid.

### Methods

Data were obtained from the National Health and Nutrition Examination Survey (NHANES). Four dietary indices were calculated: the Mediterranean Diet (MED), Healthy Eating Index-2020 (HEI-2020), Alternate Healthy Eating Index (AHEI), and Dietary Approaches to Stop Hypertension (DASH). Univariate logistic regression analysis and restricted cubic splines (RCS) curve were used to analyze the single effect of dietary index and kidney stones. Weighted Quantile Sum (WQS) and Bayesian Kernel Machine Regression (BKMR) to visualize the mixed effects of multiple dietary indices and kidney stone risk. Finally, mediation analyses were employed to assess associations and mediation effects.

### Results

Among 25,421 participants, 2,470 had a history of kidney stones. All four dietary indices showed a significant negative association with kidney stone risk, with MED and HEI-2020 showing the strongest effects. WQS regression results indicated that the mixed effects of the four dietary indices were negatively correlated with kidney stones, with the highest weight attributed to HEI2020 (49.2%), followed by DASH (26.4%), MED (21.8%), and AHEI (2.9%). BKMR analysis revealed a negative exposure-response trend for each dietary indices to the risk of kidney stones

**Data availability statement:** All relevant data are within the manuscript and its Supporting Information files.

**Funding:** This work was supported by the Natural Science Basic Research Program of Shaanxi Province (Grant No. 2020JQ-544), and the National Science Foundation for Young Scientists of China (Grant No. 82100812).

**Competing interests:** The authors have declared that no competing interests exist.

and HEI2020, DASH, MED, and AHEI are negative related with kidney stones at all three quantiles. Mediation analysis revealed that uric acid mediated the relationship between dietary indices and kidney stone risk, with mediation proportions of 25.56% (MED), 12.14% (AHEI), 5.88% (DASH), and 2.52% (HEI-2020).

## Conclusion

Healthy dietary patterns are associated with a reduced risk of kidney stones, partially mediated by uric acid levels.

---

## 1. Introduction

Kidney stone is one of the common urinary system diseases worldwide, and the prevalence is increasing worldwide. In the United States, the prevalence of kidney stones has long been around 10% in the adult population, with a higher incidence among males compared to females [1–3]. The formation of kidney stones is influenced by a variety of common factors. Underlying conditions such as obesity, gout, asthma, and diabetes can all contribute to an increased incidence of kidney stones [4–6]. Furthermore, inadequate fluid intake [7], a high-salt diet [8], diets rich in animal protein, and low-fiber dietary habits can elevate the risk of stone formation [9]. Therefore, it is important to investigate the diet-related factors associated with the occurrence of kidney stones and develop effective prevention strategies.

A healthy balanced diet is very essential to prevent kidney stones. Dietary indices are widely used tools to evaluate overall diet quality and its relation to disease. Previous studies have shown that higher Dietary Inflammatory Index (DII) scores and increased intake of pro-inflammatory diets are associated with an increased incidence and recurrence of kidney stones [10]. Niloofar et. al found that a higher Dietary Insulin Load (DIL) is directly related to the likelihood of kidney stones [11]. Additionally, a large cohort study in China indicated that adhering to a balanced dietary pattern, rather than a plant-based diet, is associated with a lower risk of kidney stones [12]. Of the main dietary indices, four well-established indices—the Mediterranean Diet (MED), the Healthy Eating Index-2020 (HEI-2020), the Alternate Healthy Eating Index (AHEI), and the Dietary Approaches to Stop Hypertension (DASH)—emphasize higher intake of fruits, vegetables, whole grains, and plant-based proteins, and lower intake of red meat, sodium, and saturated fat [13,14]. These patterns are known to improve metabolic health and reduce oxidative stress.

Uric acid is a key biochemical factor in stone formation, especially for urate and calcium-containing stones. Diet strongly influences uric acid metabolism: high-purine animal protein, fructose-rich beverages, and alcohol increase uric acid levels, whereas plant-based and low-fat diets reduce them [15]. Therefore, uric acid may act as a mediator linking dietary patterns and kidney stone risk, but the specific mechanisms underlying this process still lack adequate evidence.

Given these mechanisms, we hypothesized that healthy dietary patterns might reduce the risk of kidney stones in part through modulation of uric acid levels. Although several previous studies have investigated individual dietary indices (e.g., DASH or MED) using NHANES or cohort data [16,17], few have comprehensively compared multiple healthy dietary indices within the same population while examining uric acid as a biological mediator. This integrative approach may provide new insight into the mechanisms linking diet and stone formation. Our study uncovers potential associations between healthy dietary patterns and kidney stones, and to provide a scientific basis for the prevention and treatment of kidney stones through dietary interventions.

## 2. Methods

### 2.1 Study population and design

The raw data for this study were obtained from the NHANES III database (https://www.cdc.gov/nchs/nhanes/), with the study population and design conducted by the Centers for Disease Control and Prevention (CDC) and the National Center for Health Statistics (NCHS). The data from NHANES has been approved by the Institutional Review Board of the NCHS, and all participants provided written informed consent. We included data from six cycles (2007–2018), a total of 59,842 participants, as these cycles provided both kidney stone history and dietary and uric acid test data. Participants were excluded if they were under 18 years old and lacked kidney stone information (25,072 individuals), lacked dietary information (7,923 individuals), and lacked uric acid data (1,335 individuals). In the end, 25,421 participants were included in this study, with the detailed screening process shown in Fig 1.

### 2.2 Dietary indices

The raw data of the four dietary indices was assessed using 48-hour dietary recall data. Dietary intake data were used to estimate the types and quantities of foods and beverage consumed during the 48-hour prior to the interview and to estimate the intake of energy, nutrients, and other food components from these foods and beverages. These data were collected through two retrospective interviews. Nutrients and energy for each food or beverage were calculated using the Food and Nutrition Database for Dietary Studies (FNDDS), The United States Department of Agriculture Food Pattern Equivalents Database (FPED) guideline was used for food group classification. Our analysis is grounded in data from four widely recognized dietary pattern indices that serve as proxies for dietary quality: MED, HEI2020, AHEI and DASH diets. The MED is a healthy dietary pattern characterized by increased consumption of legumes, vegetables, fruits, olive oil, whole grain grains, and nuts. It also includes moderate consumption of fish and red wine, while limiting consumption of red meat products and saturated fatty acids. The HEI-2020 assesses diet quality according to the Dietary Guidelines for Americans 2020−2025–13 food groups or nutrient components, with a total score of 100, and higher scores indicate better diet quality. It provides a comprehensive assessment of dietary habits and the complex effects of diet on disease risk and metabolic disorders. The AHEI was extended from HEI-2010 to include more dietary factors related to chronic disease risk, based on the intake scores of 11 foods and nutrients. The DASH diet was originally developed by the National Heart, Lung, and Blood Institute (NHLBI) as a dietary model for the prevention and control of hypertension and is now considered to be one of the ideal dietary options for all adults. The four dietary indices were calculated with the use of the dietaryindex R package [18].

### 2.3 Definition of kidney stones

The Kidney Conditions – Urology questionnaire was directed at adults aged 20 years and older, which includes questions about a history of kidney stones. Survey participants who answered yes to '(KIQ026) Have you/Has sample person (SP) ever had a kidney stone?' were considered to have a history of kidney stones. And no was considered to no kidney stones history. Don't known and refused were excluded.

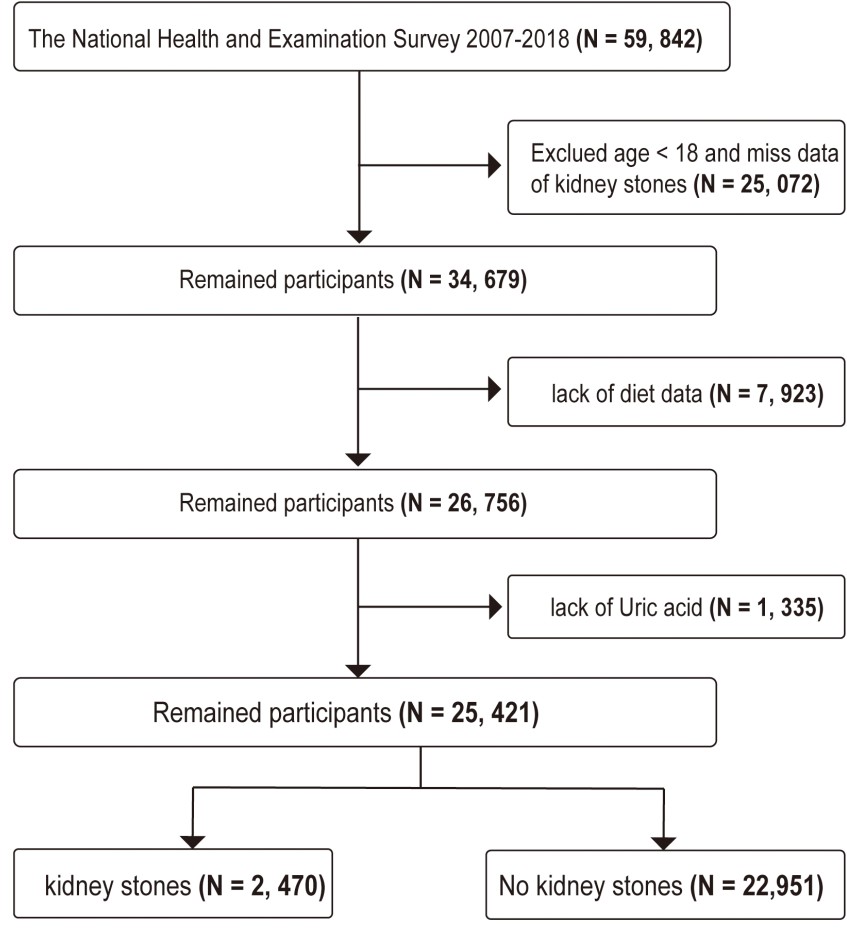

**Fig 1. Flowchart of the study.**

## 2.4 Mediation variables assessment

A standard biochemistry test was conducted by trained laboratory technicians, and uric acid concentration was measured using a timed endpoint method. Detailed instructions about analytical methodologies, principles, and operating procedures are shown in the NHANES Laboratory Method Files.

## 2.5 Covariate assessment

Potential covariates were identified a priori based on a literature review [8,10], including age, Body Mass Index (BMI), cotinine (substitute for smoking), gender (male/female), marital status (Divorced/ Living with partner/ Married/ Never married/ Separated/ Widowed), alcohol, race (Mexican American/ Non-Hispanic Black/ Non-Hispanic White/ Other Hispanic/ Other Race – Including Multi-Racial), education (College Graduate or above/ High School Grad/ Less Than 9th Grade/ Some College or AA degree), hypertension and Diabetes. Cotinine content, it would be more accurate to quantify the level of human exposure to tobacco smoke [19,20]. Alcohol status was categorized into five groups: never consumption (<12 drinks in lifetime), former consumption (≥12 drinks in 1 year and did not drink last year, or did not drink last year but drank ≥12 drinks in lifetime), mild consumption (1 drinks/d for female and 2 drinks/d for male), moderate consumption (2 drinks/d for female and 3 drinks/d for male), and heavy consumption (3 drinks/d for female and 4 drinks/d for male) [21].

## 2.6 Statistical analysis

Statistical analyses were performed according to the NHANES data analysis reporting guidelines, and all analyses were performed on a weighted treatment basis. In data processing, continuous data were expressed mean±standard deviation and analyzed by t-tests, categorical data were expressed as proportion and analyzed by χ2 test.

To assess the robustness of the associations between dietary indices and kidney stones, three logistic regression models were constructed. The crude model estimated the unadjusted association between each dietary index and kidney stone risk. Model 1 was adjusted for demographic factors, including age, sex, and race. Model 2 was further adjusted for socioeconomic and clinical variables, including education level, marital status, BMI, cotinine, alcohol consumption, hypertension, and diabetes. This stepwise modeling approach was used to determine whether the observed associations were independent of demographic and clinical confounding factors. Restricted cubic spline (RCS) models were used to determine a possible dose-response relationship between dietary index and kidney stones. The weighted quantile sum (WQS) regression model explored the association between multiple dietary indices and kidney stones. The magnitude of the weights shows the contribution of each dietary index to the WQS index, with higher values indicating greater effects. Data were tested and validated using 40% and 60% stratified random samples, with 1000 bootstrap iterations [22]. Given the potential nonlinear and non-additive relationships between dietary index scores, Bayesian kernel machine regression (BKMR) was used to visualize the association between multiple dietary index scores and kidney stone risk [23]. Finally, we evaluated the mediation effect of four dietary indices on the effect of kidney stones through uric acid.

R (version 4.2.1) was used for all statistical analyses, and P<0.05 (two-side test) were considered to statistical significance. WQS was performed using the gWQS package, BKMR was performed using the BKMR package, and mediation effect analysis was performed using the Mediation package (Table 1).

## 3. Results

### 3.1 Baseline characteristics of the participants

### 3.2 Association between dietary index and risk of kidney stones

To explore the association between individual dietary indices and the risk of kidney stones, we performed univariate logistic regression with adjustment for relevant confounders on the basis of the presence or absence of kidney stones. The results are shown in Table 2. After adjusting for confounding variables, the risk of kidney stones decreased with increasing scores on all four dietary indices, with AHEI (OR:0.992, 95% CI: 0.990–0.994), DASH (OR:0.976, 95% CI: 0.970–0.991), HEI2020 (OR:0.989, 95% CI: 0.987–0.991), MED (OR:0.941, 95% CI:0.904–0.979), all P<0.05. This suggests that a healthier diet is important for the prevention of kidney stones.

### 3.3 Non-linearity and threshold effect analysis between dietary index and risk of kidney stones

RCS curves were used to examine potential nonlinear associations in the relationship between the four dietary indices and the likelihood of kidney stones, and the results are shown in Fig 2. The overall associations between the four dietary indices (AHEI, DASH, HEI2020, and MED) and kidney stone risk were all statistically significant (all P-overall<0.001). The HEI2020 and MED with the kidney stone risk shows a linear trend (P value for Nonlinear >0.05), while the AHEI and DASH with the kidney stone risk exhibit nonlinear trends (P value for Nonlinear <0.05). Of which, AHEI displayed an inverted U-shaped association with kidney stone risk, with the inflection point occurring at AHEI index scores: 37.73. DASH exhibited an inverted U-shaped association with kidney stone risk, with the inflection point occurring at DASH index scores: 21.96.

### 3.4 Association of mixed effects of dietary index with kidney stones evaluated by WQS

We performed Pearson correlation analysis on the four dietary indices and found significant positive correlations among the four dietary indices (as shown in Fig 3A). Fig 3B show the results of the mixed effects of dietary components from various dietary indices on the risk of kidney stone within the WQS model. WQS regression results indicated that the mixed

**Table 1. Baseline characteristics of included participants.**

| Characteristics | Overall | Non-kidney stone | Kidney stone | P |
|---|---|---|---|---|
| Number | 25421 | 22951 | 2470 | |
| Age (Years) | 50.26±17.52 | 49.58±17.56 | 56.50±15.87 | <0.001 |
| Age category | | | | <0.001 |
| < 60 | 16665 (65.6%) | 15383 (67.0%) | 1283 (51.90%) | |
| ≥60 | 8756 (34.4%) | 7569 (33.0%) | 1187 (48.10%) | |
| Gender | | | | <0.001 |
| Female | 11034 (52.5%) | 10127 (53.5%) | 907 (43.8%) | |
| Male | 9967 (47.5%) | 8802 (46.5%) | 1165 (56.2%) | |
| BMI (kg/m2) | 29.48±7.01 | 29.36±7.02 | 30.52±6.77 | <0.001 |
| BMI category | | | | <0.001 |
| < 18.5 | 364 (1.4%) | 347 (1.5%) | 17 (0.7%) | |
| 18.5-24.9 | 6566 (26.1%) | 6109 (26.9%) | 457 (18.7%) | |
| 25.0-29.9 | 8290 (32.9%) | 7466 (32.8%) | 824 (33.7%) | |
| ≥30 | 9962 (39.6%) | 8818 (38.8%) | 1144 (46.8%) | |
| Race | | | | <0.001 |
| Mexican American | 3317 (15.7%) | 3055 (16.0%) | 262 (12.5%) | |
| Non-Hispanic Black | 4192 (19.8%) | 3933 (20.6%) | 259 (12.4%) | |
| Non-Hispanic White | 9317 (44.0%) | 8110 (42.5%) | 1207 (57.8%) | |
| Other Hispanic | 2211 (10.4%) | 1996 (10.5%) | 215 (10.3%) | |
| Other Race | 2126 (10.0%) | 1981 (10.4%) | 145 (6.9%) | |
| Education | | | | 0.022 |
| Less Than 9th Grade | 1997 (9.4%) | 1804 (9.5%) | 193 (9.2%) | |
| 9-11th Grade | 2841 (13.4%) | 2549 (13.4%) | 292 (14.0%) | |
| High School Grad | 4906 (23.2%) | 4431 (23.2%) | 475 (22.7%) | |
| Some College or AA degree | 6338 (29.9%) | 5654 (29.6%) | 684 (32.8%) | |
| College Graduate or above | 5060 (23.9%) | 4617 (24.2%) | 443 (21.2%) | |
| Refused | 6 (0.0%) | 6 (0.0%) | 0 (0.0%) | |
| Missing | 15 (0.1%) | 14 (0.1%) | 1 (0.0%) | |
| Marital Status | | | | <0.001 |
| Married | 11177 (52.8%) | 9961 (52.2%) | 1216 (58.2%) | |
| Living with partner | 1688 (8.0%) | 1562 (8.2%) | 126 (6.0%) | |
| Divorced | 2364 (11.2%) | 2075 (10.9%) | 289 (13.8%) | |
| Separated | 695 (3.3%) | 630 (3.3%) | 65 (3.1%) | |
| Never married | 3601 (17.0%) | 3405 (17.9%) | 196 (9.4%) | |
| Widowed | 1631 (7.7%) | 1436 (7.5%) | 195 (9.3%) | |
| Refused | 6 (0.0%) | 5 (0.0%) | 1 (0.0%) | |
| Missing | 1 (0.0%) | 1 (0.0%) | 0 (0.0%) | |
| HBP | | | | <0.001 |
| No | 13360 (63.1%) | 12351 (64.7%) | 1009 (48.3%) | |
| Yes | 7779 (36.8%) | 6702 (35.1%) | 1077 (51.6%) | |
| Missing | 24 (0.1%) | 22 (0.1%) | 2 (0.1%) | |
| Diabetes | | | | <0.001 |
| No | 17823 (84.2%) | 16276 (85.3%) | 1547 (74.1%) | |
| Borderline | 520 (2.5%) | 454 (2.4%) | 66 (3.2%) | |
| Yes | 2809 (13.3%) | 2336 (12.2%) | 473 (22.7%) | |

*(Continued)*

**Table 1.** (Continued)

| Characteristics | Overall | Non-kidney stone | Kidney stone | P |
|---|---|---|---|---|
| Missing | 11 (0.1%) | 9 (0.0%) | 2 (0.1%) | |
| Cotinine (ng/mL) | 55.57±127.20 | 54.49±125.57 | 65.39±140.91 | <0.001 |
| Alcohol | | | | <0.001 |
| Never | 4174 (19.9%) | 3771 (19.9%) | 403 (19.4%) | |
| Former | 1020 (4.9%) | 911 (4.8%) | 109 (5.3%) | |
| Mild | 7900 (37.6%) | 7040 (37.2%) | 860 (41.5%) | |
| Moderate | 3673 (17.5%) | 3334 (17.6%) | 339 (16.4%) | |
| Heavy | 4234 (20.2%) | 3873 (20.5%) | 361 (17.4%) | |
| MED | 3.47±1.37 | 3.48±1.37 | 3.37±1.34 | <0.001 |
| HEI2020 | 51.52±12.04 | 51.64±12.09 | 50.37±11.54 | <0.001 |
| AHEI | 38.53±11.49 | 38.59±11.53 | 38.02±11.07 | 0.032 |
| DASH | 22.37±5.01 | 22.40±5.04 | 22.08±4.80 | 0.005 |
| Uric acid (mg/dL) | 5.45±1.45 | 5.43±1.44 | 5.67±1.52 | <0.001 |

Mean±SD for continuous variables, % for categorical variables, The two columns of data, Age (Years) and BMI (kg/m$^2$), have the minimum and maximum values indicated within the brackets. BMI, body mass index; HBP, hypertension; MED, Mediterranean Diet; HEI2020, Healthy Eating Index-2020; AHEI, Alternate Healthy Eating Index; DASH, Dietary Approaches to Stop Hypertension.

effects of the four dietary indices were negatively correlated with kidney stones, with the highest weight attributed to HEI2020 (49.2%), followed by DASH (26.4%), MED (21.8%), and AHEI (2.9%).

### 3.5 BKMR model to assess the effect of dietary index on kidney stones

We used the BKMR model, which is a nonparametric Bayesian variable selection framework to assess the combined effect of multiple dietary indices on kidney stones. We initially assessed the univariate exposure-response associations between single dietary index and kidney stones. Fig 3C illustrates a negative exposure-response trend for each dietary indices to the risk of kidney stones across the total population. The overall effect of the BKMR mixture assessment indicated that dietary indices levels at the 60th percentile and above were associated with significantly lower rates of kidney stones than exposure levels fixed at the 50th percentile (Fig 3D). Additionally, Fig 3E illustrates the single-variable effects of various dietary indices on the risk of kidney stones at the 25th, 50th, and 75th quantiles. HEI2020, DASH, MED, and AHEI are negative related with kidney stones at all three quantiles, indicating that higher levels of these dietary indices are correlated with a decreased risk of kidney stones.

The bivariate relationship was further explored by examining four dietary indices. Fig 4 presents the relationship between expos1 and the quantiles of expos2 for various dietary indices scores (HEI2020, DASH, MED, and AHEI). It

**Table 2.** Multiple logistic regression analysis between four dietary indices and kidney stones.

| Variable | Crude model | | Model 1 | | Model 2 | |
|---|---|---|---|---|---|---|
| | OR [95% CI] | p-value | OR [95% CI] | p-value | OR [95% CI] | p-value |
| AHEI | 0.995 (0.993, 0.997) | <0.001 | 0.988 (0.986, 0.990) | 0.001 | 0.992 (0.990, 0.994) | 0.003 |
| DASH | 0.985 (0.975, 0.995) | <0.001 | 0.965 (0.957, 0.974) | <0.001 | 0.976 (0.970, 0.982) | <0.001 |
| HEI2020 | 0.990 (0.988, 0.992) | <0.001 | 0.984 (0.982, 0.986) | <0.001 | 0.989 (0.987, 0.991) | <0.001 |
| MED | 0.929 (0.863, 0.999) | <0.001 | 0.904 (0.870, 0.939) | <0.001 | 0.941 (0.904, 0.979) | 0.001 |

Crude model: no covariates were adjusted.

Model 1: sex, age, race, educational level, and marital status were adjusted.

Model 2: model 1＋BMI, Cotinine, Alcohol drink, Diabetes and Hypertension (HBP) were fully adjusted.

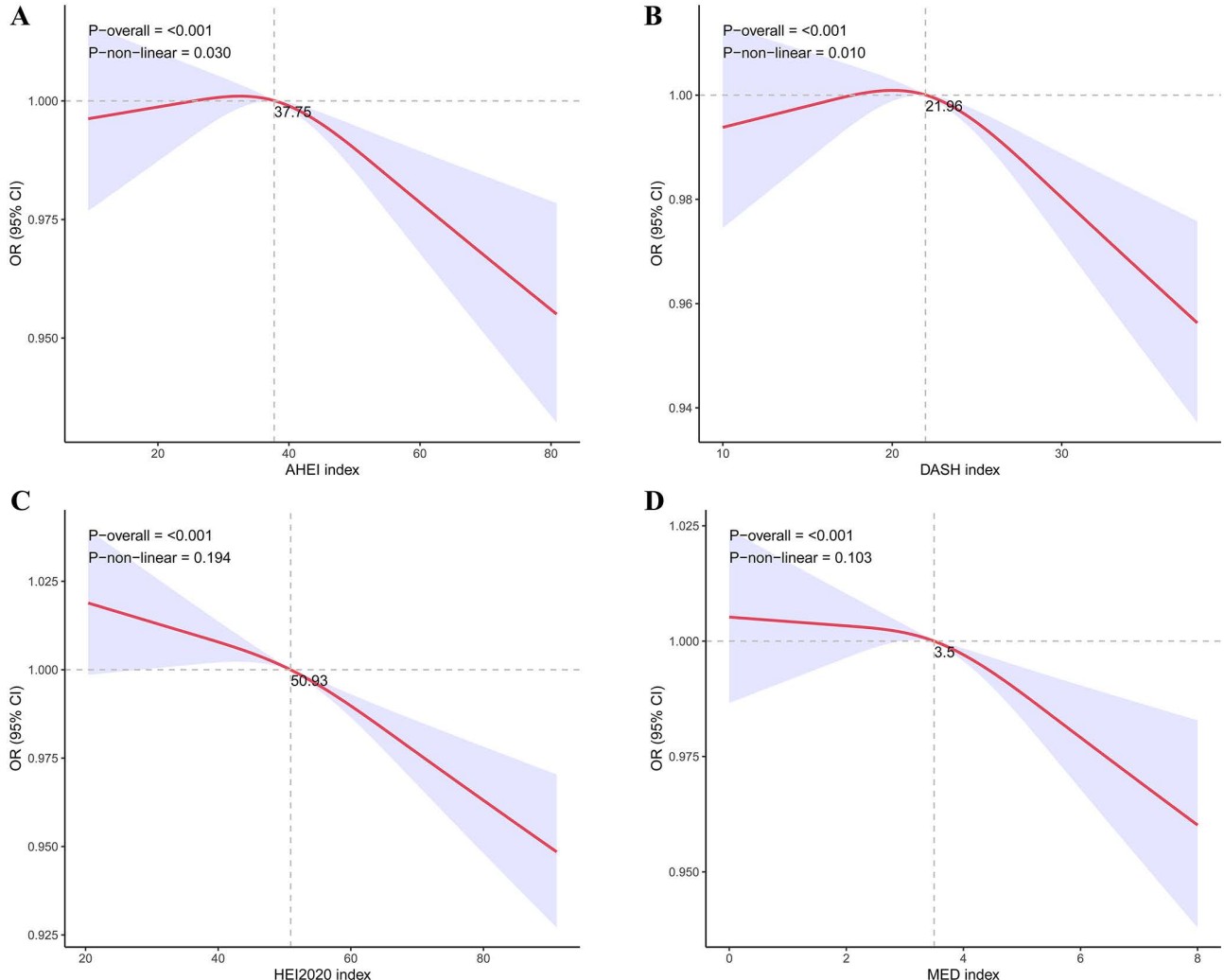

**Fig 2. The exposure-response association of the dietary indices with the kidney stone by restricted cubic spline (RCS).** (A) Nonlinear association between AHEI index and the risk of kidney stone. (B) Nonlinear association between DASH index and the risk of kidney stone. (C) Nonlinear association between HEI2020 index and the risk of kidney stone. (D) Nonlinear association between MED index and the risk of kidney stone. OR: odds ratio.

shows the relationship between dietary indices in the column and kidney stones when the dietary index mixture in the row is fixed at its 25th, 50th, and 75th percentiles and the remaining dietary indices are fixed at its 50th percentile. There was a clear interaction between these dietary indices.

### 3.6 Intermediation effect analysis

As shown in the weighted multivariable linear regression models (Table 3), higher scores on all four healthy dietary indices were significantly associated with lower serum uric acid concentrations (all P < 0.05), with the inverse associations for MED and HEI-2020 being the most pronounced. These findings support the role of uric acid as a potential mediator in the relationship between dietary patterns and kidney stones. Further mediation analysis (Fig 5) indicated that uric acid was closely associated with all four dietary indices and acted as a mediator in their relationships with kidney stones. The mediating proportions of AHEI, DASH, HEI2020 and MED were 12.14%, 5.88%, 2.52% and 25.56%, respectively. The P value

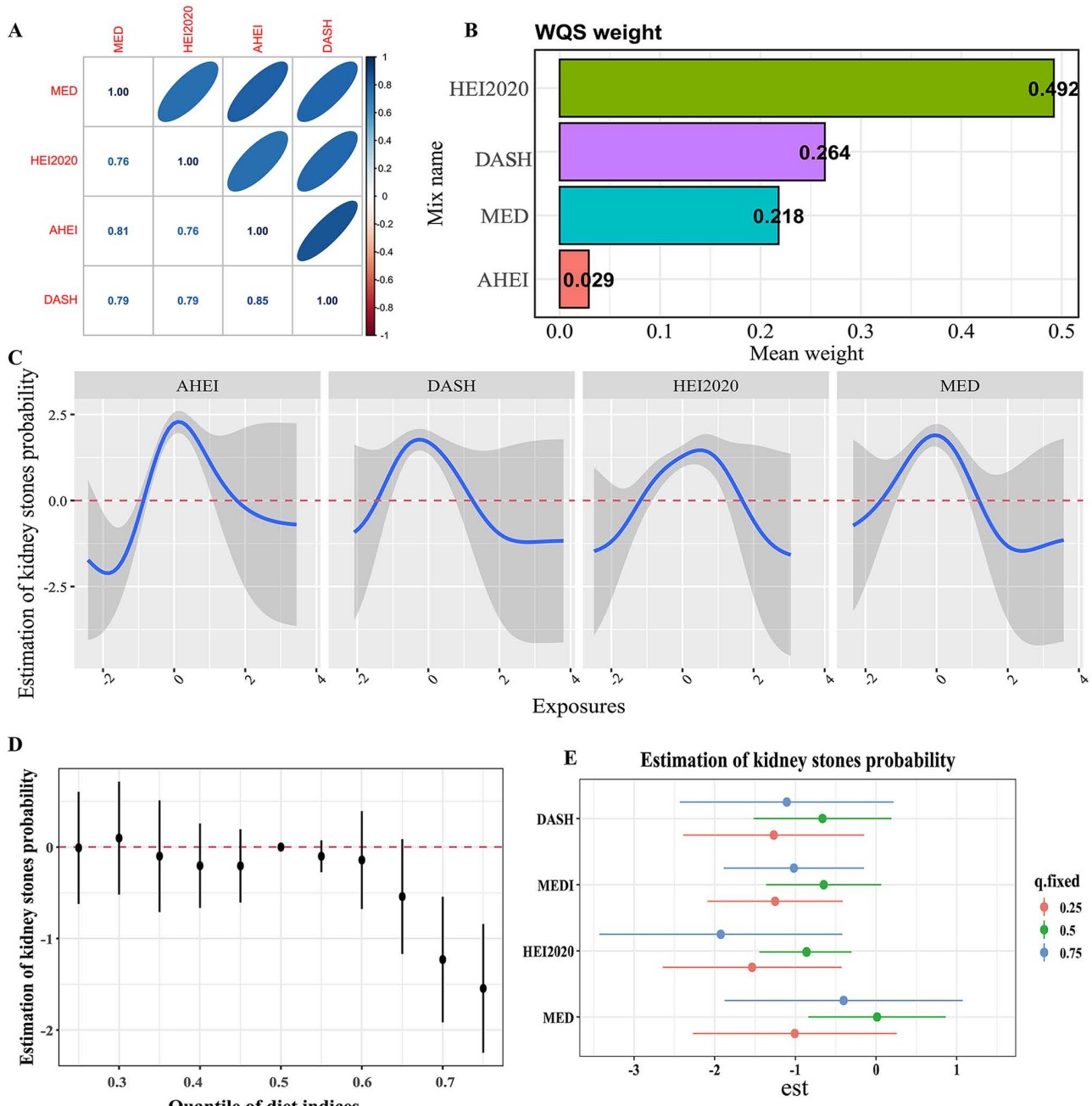

**Fig 3. Mixed effects and total effects of four dietary indices on kidney stone.** (A) Spearman correlations among the MED, DASH, AHEI, and HEI-2020. (B) Mixed effects of the four dietary indices on kidney stone assessed by WQS. (C) Univariate exposure–response function and 95% CI scores for the associations between single dietary indices exposures when other dietary indices exposures are fixed at the median level. (D) Summary of the overall health effects of the combined exposures on the four dietary indices at various quantiles (from 25th to 75th) by BKMR. (E) Single-variable effects of dietary indices at decreasing quartiles for the kidney stone.

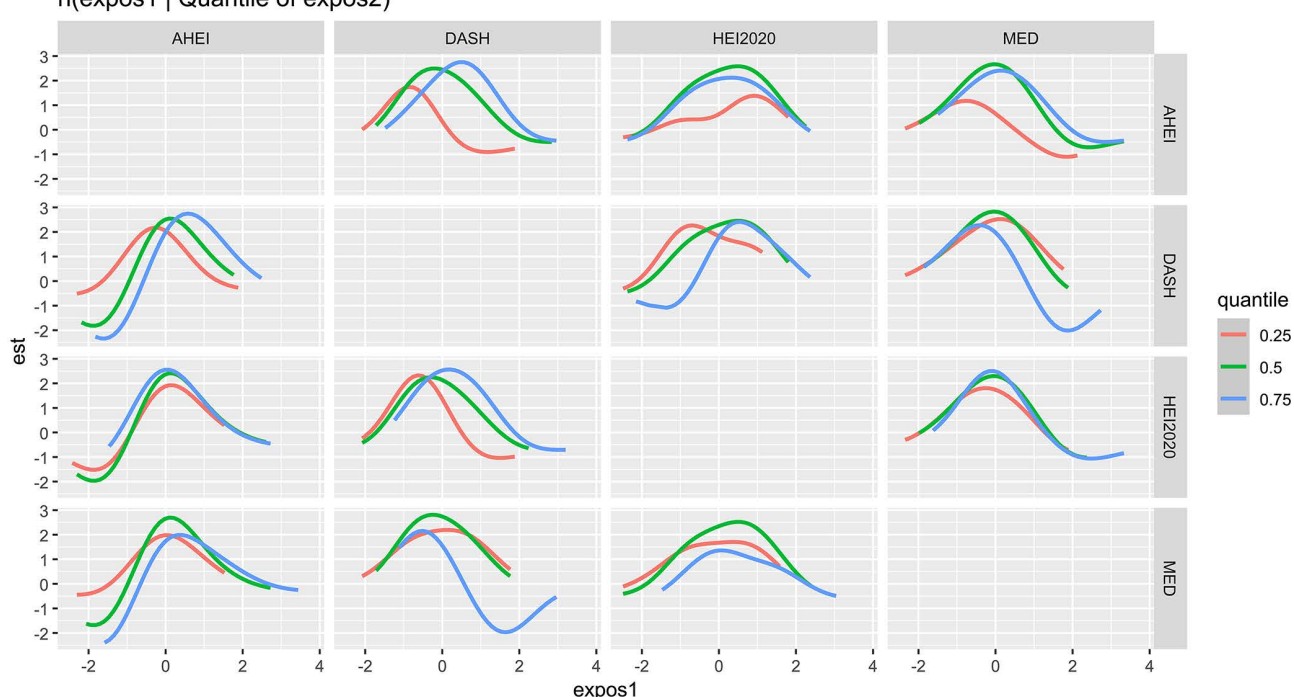

**Fig 4. Bivariate exposure–response functions of dietary indices with the kidney stone.**

**Table 3. The linear regression results of four dietary indices and uric acid.**

| Dietary indices | β(95%CI) | t | P |
|---|---|---|---|
| MED | −1.5241(−2.644, −0.405) | −2.668 | 0.010 |
| HEI2020 | −1.179 (−1.882, −0.476) | −3.287 | 0.001 |
| AHEI | −0.251(−0.375, −0.127) | −3.962 | < 0.001 |
| DASH | −0.824(−1.182, −0.467) | −4.517 | < 0.001 |

Linear regression models were adjusted for age, gender, race, education, BMI, cotinine, hypertension, and diabetes. MED, Mediterranean Diet; HEI2020, Healthy Eating Index-2020; AHEI, Alternate Healthy Eating Index; DASH, Dietary Approaches to Stop Hypertension.

of mediating effect of DASH was 0.062, and the rest were less than 0.05, indicating that healthy diet had a great impact on the risk of kidney stones, and there was a mediating effect mediated by uric acid.

## 4. Discussion

This study investigated the association between the four dietary indices and the risk of kidney stones and the mediating role of uric acid levels in the relationship between diet and kidney stones. By analyzing the data from the NHANES database, we found that the OR for kidney stones decreased significantly with increasing AHEI, DASH, HEI2020, MED. This negative relationship remained robust after adjusting for potential confounders such as age, sex, race, Poverty Income Ratio (PIR), education level, marriage, BMI, smoking, alcohol consumption, Diabetes Mellitus (DM), and hypertension.

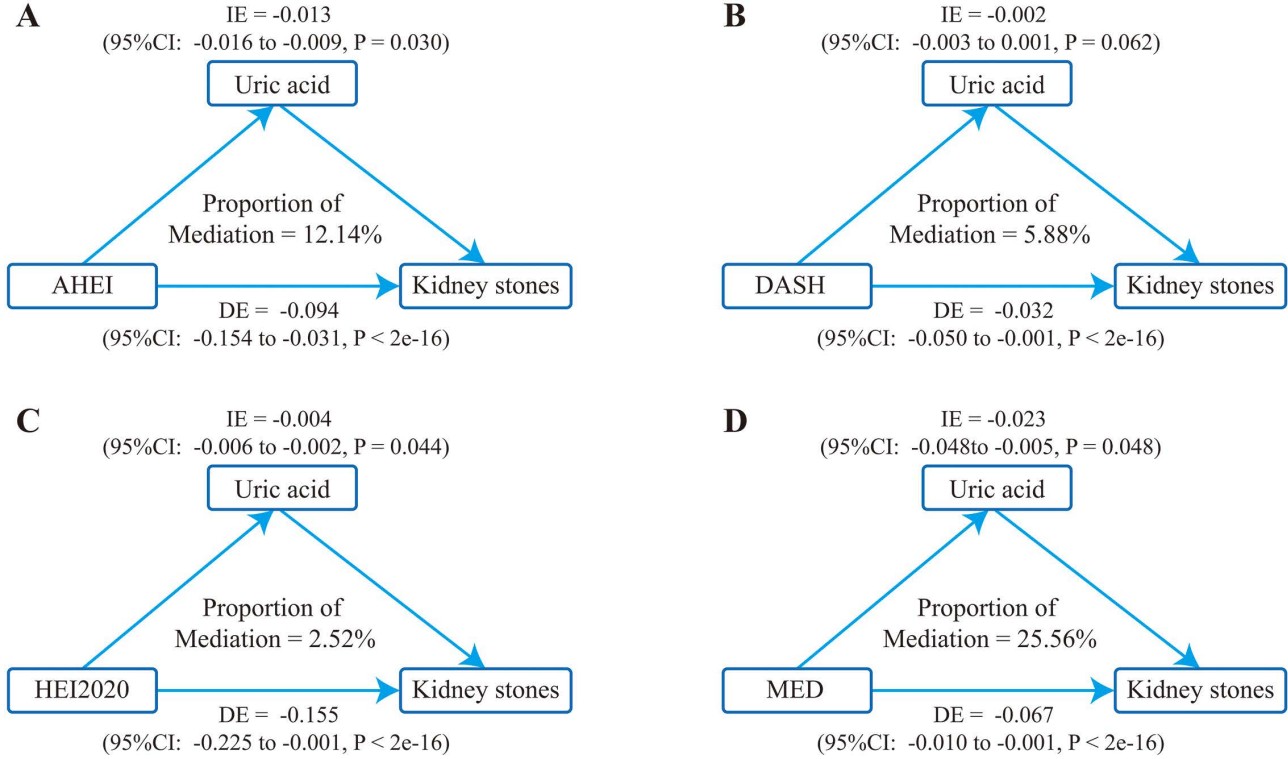

**Fig 5. Uric acid mediates the associations between dietary indices and the risk of kidney stone.** (A) Partial mediating effect of uric acid on the association between AHEI and the risk of kidney stone. (B) DASH and kidney stone. (C) HEI2020 and kidney stone. (D) MED and kidney stone. DE: Direct effect; IE: Indirect effect.

Specifically, AHEI and DASH were inversely associated with the risk of kidney stones only above 31.75 and 21.96, while HEI2020 and MED were inversely associated with the risk of kidney stones. In addition, blood uric acid levels mediated the four dietary indices and the risk of kidney stones.

Multiple analyses were performed for the separate effects of the four dietary indices on the risk of kidney stones. Logistic regression analysis showed that MED showed a lower OR value in Crude model, Model 1 and Model 2. A stepwise modeling approach was used to demonstrate the stability of associations. The persistence of significant effects across Crude, Model 1, and Model 2 suggests that the observed relationships were not driven solely by demographic or clinical confounding. Correlation analysis showed that there was a high correlation among the four dietary indices, and the WQS regression model could be used to explore the total effect of mixed exposure and the relative contribution of each factor. WQS weight showed that HEI2020 had the highest proportion weight of 0.492, that is, HEI2020 had the greatest impact on the risk of kidney stones. In the BKMR forest plot, HEI2020 indicated a lower risk of kidney stones when quartile detection of the index was performed. In conclusion, in separate effect analyses, the MED and HEI2020 indexes had the greatest impact on kidney stone risk.

The AHEI advocates the consumption of fruits, vegetables, grains, and vegetable oils, while discouraging the consumption of sugar-sweetened beverages, alcohol, and meat [24]. AHEI was found to be significantly associated with gallstone risk reduction in previous studies [25], furthermore, the AHEI index has been proven to cause kidney stones through biological aging [26]. In the present study, we found that AHEI diet can inhibit the occurrence of kidney stones when it exceeds 31.75.

The DASH diet has been associated with a reduction in kidney stones in previous studies [27]. The DASH diet is associated with decreased calcium oxalate supersaturation, increased magnesium and citrate excretion, and increased urine pH [28]. Higher DASH score was associated with higher urine potassium, magnesium, phosphate, and pH, and lower relative supersaturations (RSS) of calcium oxalate (women only) and uric acid [29]. The DASH diet reduced the incidence of stones by changing the characteristics of urine [30]. The DASH diet is characterized by a high intake of fruits, vegetables, whole grains, and low-fat dairy products, as well as a low intake of sodium, red and processed meats, and added sugars. The DASH recommends a reduction in sodium intake while emphasizing dairy intake to promote kidney health and is related to the intake of calcium and phosphorus [31]. The DASH diet, which restricts the intake of meat protein, may reduce stone formation to a large extent through the mediating effect of uric acid.

HEI2020 encourages high intake of vegetables, fruits, and low-fat dairy products, which help to reduce uric acid levels and increase urine pH. Limiting refined grains and foods with a high oxalate content may reduce the risk of calcium oxalate stones. Emphasizing a balanced intake of calcium and magnesium in the diet can help reduce the likelihood of stone formation. Diet may influence stone risk by affecting the gut microbiota and altering oxalate metabolism and absorption [32,33]. This is consistent with the inhibition of stone formation by HEI2020 in this study.

The MED Mediterranean diet is rich in potassium, magnesium, and low in saturated fat, components that help to increase urine pH and reduce urinary calcium excretion, thereby reducing the risk of calcium oxalate and urate stones. A high-potassium dietary pattern is associated with a reduced risk of kidney stones, and fruits and vegetables in the Mediterranean diet are the main sources of potassium [34,35]. Kidney stone formation may be associated with chronic inflammation and oxidative stress, and the antioxidant components such as polyphenols and anti-inflammatory mechanisms of the Mediterranean diet help reduce these risks [36].

Because the healthy eating index was generally consistent in terms of dietary types, there was a high correlation and interaction among the four diets in this study, and the combined effect of multiple healthy eating indexes was significantly higher in reducing the risk of kidney stones. The diversity of healthy dietary patterns may optimize the urinary metabolic environment through the combined intake of potassium, magnesium, calcium, and antioxidant components. Reducing urinary calcium excretion (e.g., DASH), regulating urine pH (e.g., MED), and lowering uric acid levels (e.g., AHEI) may reduce stone risk through a combination of different pathways [37,38]. A study based on the NHS in the United States showed that the DASH and MED diets reduced the incidence of kidney stones, respectively, and this protective effect may be achieved by regulating urinary metabolic components and reducing inflammation [39]. In addition, this study identified lower uric acid levels as a mediator between multiple healthy dietary states and the risk of kidney stones. Uric acid level is an important cause of renal uric acid stones and other types of stones [40] which is affected by healthy diet level [41]. The AHEI diet emphasizes reducing the consumption of sugar-sweetened beverages, a feature that is particularly important for controlling uric acid levels [42]. The DASH diet was associated with most metabolic parameters, negatively correlated with triglycerides, urinary sodium, and uric acid, and positively correlated with serum vitamin D [43, 44]. Olive oil and nuts of the MED diet are rich in polyphenolic antioxidants, which reduce uric acid production and related tissue damage by inhibiting inflammation and oxidative stress. Both DASH and MED diets contribute to weight management, improve insulin resistance, and indirectly reduce uric acid levels [45,46].

Despite the significant findings of this study, several limitations should be acknowledged. First, the cross-sectional nature of the analysis precludes establishing causal relationships. Future longitudinal studies are essential to confirm the observed associations and mediation effects. Second, dietary data were based on 24-hour recall interviews, which may introduce recall bias and fail to capture long-term dietary habits accurately. Third, although uric acid's mediating role was identified, the precise molecular mechanisms underlying this mediation remain unexplored and warrant further biological investigation. Fourth, while extensive covariate adjustments were performed, residual confounding due to unmeasured or unknown factors cannot be entirely ruled out. The strengths of this study include the use of a large, nationally representative sample with standardized dietary and biochemical assessments, the simultaneous evaluation of four major dietary

indices, and the application of advanced analytical models (WQS and BKMR). From a public health perspective, these findings highlight that promoting healthy dietary patterns may help reduce kidney stone risk, partly through the regulation of uric acid metabolism. Moreover, artificial intelligence–driven dietary applications may offer innovative tools for personalized dietary monitoring and guidance, further supporting kidney stone prevention through improved nutritional balance.

## 5. Conclusion

In conclusion, this study provides robust evidence linking healthy dietary patterns to reduced kidney stone risk. Four dietary indices, including MED, DASH, AHEI, and HEI-2020, were negatively associated with kidney stone prevalence. MED and HEI-2020 had the strongest protective effects. Mediation analysis confirmed that uric acid partially mediates these associations, with MED showing the highest mediation proportion. These findings highlight the critical role of diet in kidney stone prevention and the potential mechanisms involving uric acid regulation.

## Supporting information

**S1 Table. Distribution of dietary indices for NHANES 2007–2018 (n = 25421).**
(DOCX)

**S1 Checklist. STROBE checklist v4 combined.**
(DOCX)

## Author contributions

**Conceptualization:** Jinlong Cao, Delai Fu.

**Data curation:** Jinlong Cao, Tianyuan Zhai, Lingyu Guo.

**Formal analysis:** Jinlong Cao, Lingyu Guo, Delai Fu, Feng Li.

**Funding acquisition:** Feng Li.

**Investigation:** Tianyuan Zhai, Lingyu Guo, Qi Chen.

**Methodology:** Tianyuan Zhai, Lingyu Guo, Yue Chong, Qian Wang, Delai Fu, Feng Li.

**Resources:** Qian Wang, Feng Li.

**Software:** Yue Chong, Qi Chen, Qian Wang, Delai Fu, Feng Li.

**Supervision:** Yue Chong, Li Xue, Feng Li.

**Validation:** Yue Chong, Qi Chen, Qian Wang, Li Xue.

**Visualization:** Li Xue.

**Writing – original draft:** Li Xue.

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
