## [Decision Letter · Decision Letter 0]

15 Sep 2025

PONE-D-25-41276Uric acid levels mediate the association between four dietary indices and kidney stones in US adults: a cross-sectional study of NHANES 2007-2018PLOS ONE

Dear Dr. cao,

Thank you for submitting your manuscript to PLOS ONE. After careful consideration, we feel that it has merit but does not fully meet PLOS ONE’s publication criteria as it currently stands. Therefore, we invite you to submit a revised version of the manuscript that addresses the points raised during the review process.

Dear Authors, 

The manuscript has received an overall positive feedback from all the designated reviewers, however, many suggestions have been received to refine the clarity of the derived messages as well as to establish the validity of the inferences. Kindly go through the comments provided below and refine accordingly for further consideration. 

We look forward to receiving your revised manuscript.

Kind regards,

Yogesh Kumar Jain, MPH

Academic Editor

PLOS ONE

Journal Requirements:

https://www.researchgate.net/publication/260294589_Urinary_Lithogenic_Risk_Profile_in_Recurrent_Stone_Formers_With_Hyperoxaluria_A_Randomized_Controlled_Trial_Comparing_DASH_Dietary_Approaches_to_Stop_Hypertension-Style_and_Low-Oxalate_Diets?

https://www.mdpi.com/2072-6643/16/14/2248?

In your revision ensure you cite all your sources (including your own works), and quote or rephrase any duplicated text outside the methods section. Further consideration is dependent on these concerns being addressed.

“This work was supported by the Natural Science Basic Research Program of Shaanxi Province (Grant No. 2020JQ-544), and the National Science Foundation for Young Scientists of China (Grant No. 82100812).”

“This work was supported by the Natural Science Basic Research Program of Shaanxi Province (Grant No. 2020JQ-544), and the National Science Foundation for Young Scientists of China (Grant No. 82100812).”

6. PLOS requires an ORCID iD for the corresponding author in Editorial Manager on papers submitted after December 6th, 2016. Please ensure that you have an ORCID iD and that it is validated in Editorial Manager. To do this, go to ‘Update my Information’ (in the upper left-hand corner of the main menu), and click on the Fetch/Validate link next to the ORCID field. This will take you to the ORCID site and allow you to create a new iD or authenticate a pre-existing iD in Editorial Manager.

Reviewers' comments:

Reviewer's Responses to Questions

**Comments to the Author**

1. Is the manuscript technically sound, and do the data support the conclusions?

Reviewer #1: Partly

Reviewer #2: Yes

Reviewer #3: Yes

2. Has the statistical analysis been performed appropriately and rigorously? 

Reviewer #1: N/A

Reviewer #2: Yes

Reviewer #3: No

3. Have the authors made all data underlying the findings in their manuscript fully available?

Reviewer #1: Yes

Reviewer #2: Yes

Reviewer #3: Yes

4. Is the manuscript presented in an intelligible fashion and written in standard English?

Reviewer #1: No

Reviewer #2: Yes

Reviewer #3: Yes

5. Review Comments to the Author

Reviewer #1: Content-related Issues

1.Consider removing abbreviations that appear only once throughout the manuscript and using the full terms instead, to enhance readability.

2.Please report the observational study results following the STROBE checklist and provide the STROBE checklist in the supplementary materials.

3.The Introduction is insufficiently focused. Excessive space is devoted to describing other dietary indices, while the four dietary indices used in this study are underrepresented. This may impede reader comprehension.

4.In the Introduction, the sentence “This indicates that the animal-based diets may mediate the occurrence of kidney stones by influencing uric acid levels” appears abrupt. The preceding text does not clarify the relationship between animal-based diets and oxalate or calcium salts. Moreover, oxalate and calcium salts differ, as oxalate are primarily found in plants, not in animal-based diets.

5.Although the Introduction briefly mentions the association between uric acid and kidney stone formation, the link between dietary factors and uric acid is not addressed. Therefore, it is difficult to justify the hypothesis that uric acid mediates the relationship between dietary factors and kidney stones.

6.The statement “To our knowledge, few studies have investigated the relationship between healthy dietary indies” seems insufficiently supported. A search conducted in PubMed reveals multiple related studies, some using one of the four dietary indices, the NHANES database, or cohort study designs providing more robust evidence. The Introduction does not clearly highlight the novelty of this study, which may lead readers to question its significance. Please substantiate the rationale with adequate literature, clarify the research gap, and indicate the study’s innovation. (Note: indies is a spelling error; correct to indices.)

7. In the Methods section, the description in 2.1 Study Population and Design conflicts with the numbers and content shown in Figure 1. Please verify carefully.

8. Regarding kidney stone history in the Methods, is self-reported history sufficient to define kidney stones? How is potential bias addressed? Is there supporting evidence from high-quality literature?

9. In the Results section, based on the baseline characteristics table, the following conclusions may not be directly supported. Please reconsider and reference prior studies to accurately describe the study population:

“The smoking index cotinine indicates that smoking promotes the occurrence of kidney stones. In addition, education level and marital status also had a certain impact on kidney stones.”

10.Before conducting mediation analysis in the Results section, please consider presenting the relationship between the four dietary indices and uric acid, to justify uric acid as a potential mediator.

11.The sentence “Among them, MED restricted the intake of fat and protein more strictly, and the mediating effect of uric acid was more prominent” provides a speculative explanation and is not appropriate for the Results section. Consider moving it to the Discussion or removing it.

12.In the Discussion, the statement “However, no study has explored the relationship between AHEI and kidney stones” may be misleading. To my knowledge, prior studies have investigated the association between AHEI and kidney stones. Please review the literature carefully.

13.Please discuss the strengths of this study and its public health significance in the Discussion.

14.The figure legends lack essential information, such as covariates adjusted in the models and explanations of abbreviations in the figures. This may affect reader comprehension. Please provide the missing details.

Formatting-related Issues

15.The use of spacing throughout the manuscript is inconsistent. Please check carefully to maintain uniform formatting.

16.The references contain multiple inconsistencies in title capitalization and journal abbreviations. Please carefully review and correct these issues.

Reviewer #2: Notes

1. Did the authors check the normality of the data?

2. Can dietary apps based on artificial intelligence help reduce the risk of kidney stones? Adding this discussion point can be helpful in managing individual diets.

Reviewer #3: The study is actually interecting but it is need some revision as:

1. please explain the reason authors needs to divide the model into crude, 1 and 2

2. in the table 1 It would be more presentable if the author put age range

3. in the table 1 it would be more presentable if the author put BMI range

4. Please provides high-quality in every figures

6. PLOS authors have the option to publish the peer review history of their article (what does this mean? ). If published, this will include your full peer review and any attached files.

**Do you want your identity to be public for this peer review?** For information about this choice, including consent withdrawal, please see our Privacy Policy .

Reviewer #1: No

Reviewer #2: No

Reviewer #3: **Yes: ** Besut Daryanto

---

## [Author Response · Author response to Decision Letter 1]

15 Oct 2025

Dear Editors and Reviewers:

Thank you for your letter and for the reviewers’ comments concerning our manuscript entitled “Uric acid levels mediate the association between four dietary indices and kidney stones in US adults: a cross-sectional study of NHANES 2007-2018” (ID: PONE-D-25-41276). Those comments are all valuable and very helpful for revising and improving our manuscript, as well as the important guiding significance to our researches. We have studied comments carefully and have made correction which we hope meet with approval. Revised portion are marked in red in the paper. The main corrections in the paper and the responds to the reviewer’s comments are as flowing:

Responds to the reviewer’s comments:

Reviewer #1:

Comment 1: Consider removing abbreviations that appear only once throughout the manuscript and using the full terms instead, to enhance readability.

Response: We appreciate the reviewer’s valuable suggestion. Following your advice, abbreviations that appeared only once have been replaced with their full terms to improve clarity and readability. Specifically, “USDA” was replaced with “United States Department of Agriculture.” In addition, the following abbreviations were defined at their first appearance in the text to ensure consistency: Poverty Income Ratio (PIR), Diabetes Mellitus (DM), and Body Mass Index (BMI). These revisions have been incorporated throughout the manuscript.

Comment 2: Please report the observational study results following the STROBE checklist and provide the STROBE checklist in the supplementary materials.

Response: We thank the reviewer for this important suggestion. In accordance with the recommendation, we have carefully reviewed our manuscript to ensure compliance with the STROBE (Strengthening the Reporting of Observational Studies in Epidemiology) guidelines. A completed STROBE checklist has been prepared and included in the supplementary materials to enhance the transparency and reporting quality of our study.

Comment 3: The Introduction is insufficiently focused. Excessive space is devoted to describing other dietary indices, while the four dietary indices used in this study are underrepresented. This may impede reader comprehension.

Response: We appreciate the reviewer’s comment. The Introduction has been revised to focus more specifically on the four dietary indices analyzed in this study (MED, HEI-2020, AHEI, and DASH). Descriptions of unrelated indices (e.g., DII, DIL) were shortened, and additional details on the components and relevance of the four indices have been added to improve clarity and reader comprehension.

Comment 4: In the Introduction, the sentence “This indicates that the animal-based diets may mediate the occurrence of kidney stones by influencing uric acid levels” appears abrupt. The preceding text does not clarify the relationship between animal-based diets and oxalate or calcium salts. Moreover, oxalate and calcium salts differ, as oxalate are primarily found in plants, not in animal-based diets.

Response: We agree and have revised the paragraph to clarify the biochemical relationships between diet, uric acid, and stone composition. The new text explains that animal-based protein intake increases uric acid production, whereas oxalate is mainly derived from plant-based foods, thereby resolving the logical inconsistency.

Comment 5: Although the Introduction briefly mentions the association between uric acid and kidney stone formation, the link between dietary factors and uric acid is not addressed. Therefore, it is difficult to justify the hypothesis that uric acid mediates the relationship between dietary factors and kidney stones.

Response: Thank you for this suggestion. We added a paragraph describing how dietary components influence uric acid metabolism—highlighting the effects of purine-rich foods, fructose, alcohol, and plant-based diets—thus providing a clear rationale for investigating uric acid as a mediator.

Comment 6: The statement “To our knowledge, few studies have investigated the relationship between healthy dietary indies” seems insufficiently supported. A search conducted in PubMed reveals multiple related studies, some using one of the four dietary indices, the NHANES database, or cohort study designs providing more robust evidence. The Introduction does not clearly highlight the novelty of this study, which may lead readers to question its significance. Please substantiate the rationale with adequate literature, clarify the research gap, and indicate the study’s innovation. (Note: indies is a spelling error; correct to indices.)

Response: Thank you for this suggestion. We revised this statement to “Although several studies have examined individual dietary indices such as DASH or MED, few have comprehensively compared multiple healthy dietary indices while examining uric acid as a mediator.” We also emphasized the novelty of our integrative approach using NHANES data and clarified the research gap in the Introduction. The term “indies” was corrected to “indices.”

Comment 7: In the Methods section, the description in 2.1 Study Population and Design conflicts with the numbers and content shown in Figure 1. Please verify carefully.

Response: Thank you for identifying this inconsistency. We carefully re-checked the participant selection and confirm that the sample sizes reported in the analyses are correct. The discrepancy arose from a typographical error in the Methods text, which incorrectly stated exclusion of participants aged <20 years. We have corrected the Methods section so that it now consistently states exclusion of participants <18 years to match Figure 1. No participant counts, analytic procedures, or results were changed. We apologize for the oversight and appreciate the reviewer’s careful reading.

Comment 8: Regarding kidney stone history in the Methods, is self-reported history sufficient to define kidney stones? How is potential bias addressed? Is there supporting evidence from high-quality literature?

Response: We thank the reviewer for this valuable comment. We acknowledge that self-reported kidney stone history may be subject to potential misclassification bias. However, in large-scale epidemiological surveys such as NHANES, clinical verification for all participants is not feasible. The use of a self-reported, physician-diagnosed history collected via standardized questionnaires administered by trained interviewers is the established and widely accepted approach in this field. This standardized procedure minimizes variability.

We reviewed a large number of analyses by others on the data of NHENES kidney stones, but none of them had a process for dealing with this bias. The main references are as follows:

[1]. Chen Y, Zhang J, Li Z, Zhan Y, Tang Z, Wang J, He Z, Tang F. Causal relationship between basal metabolic rate and kidney stone disease: from discovery in US NHANES to evidence in UK Biobank cohorts. Int J Surg. 2025 Sep 1;111(9):6063-6074. doi: 10.1097/JS9.0000000000002658.

[2]. Zhou Y, Li X, He Q, Feng Q, Liu Y, Liao B. The advanced lung cancer inflammation index as a predictor of kidney stone risk in men: a cross-sectional analysis. Front Nutr. 2025 Jul 24;12:1568427. doi: 10.3389/fnut.2025.1568427.

[3]. Wei C, Yang Q, He J, Luo Y, Han K, Li J, Su S, Zhang J, Wang H, Wang D. Healthy dietary patterns, biological aging, and kidney stones: evidence from NHANES 2007-2018. Front Nutr. 2025 Mar 25;12:1538289. doi: 10.3389/fnut.2025.1538289.

Comment 9: In the Results section, based on the baseline characteristics table, the following conclusions may not be directly supported. Please reconsider and reference prior studies to accurately describe the study population:

“The smoking index cotinine indicates that smoking promotes the occurrence of kidney stones. In addition, education level and marital status also had a certain impact on kidney stones.”

Response: We thank the reviewer for this helpful observation. We agree that the original statements could be interpreted as implying causality, which is not appropriate for descriptive statistics. Accordingly, we have revised the text to present a neutral and data-driven description of the baseline characteristics. And the changes have been incorporated into the Results section of the revised manuscript. The revised statement now reads as follows: “Participants with kidney stones had higher serum cotinine levels, suggesting greater tobacco exposure, consistent with previous reports linking smoking to metabolic disturbances related to stone risk. Differences in education and marital status were also observed but require further investigation.”

Comment 10: Before conducting mediation analysis in the Results section, please consider presenting the relationship between the four dietary indices and uric acid, to justify uric acid as a potential mediator.

Response: We thank the reviewer for this valuable suggestion. In response, we added a supplementary analysis to examine the associations between each dietary index and serum uric acid levels. The results showed that higher MED, AHEI, HEI-2020, and DASH scores were significantly associated with lower uric acid concentrations (all P < 0.05), supporting the role of uric acid as a potential mediator. This new analysis has been added to the Results section and presented prior to the mediation models (Figure 5).

Comment 11: The sentence “Among them, MED restricted the intake of fat and protein more strictly, and the mediating effect of uric acid was more prominent” provides a speculative explanation and is not appropriate for the Results section. Consider moving it to the Discussion or removing it.

Response: We sincerely thank the reviewer for this valuable suggestion. We fully agree that this statement was speculative and not appropriate for inclusion in the Results section. Accordingly, we have removed this sentence from the revised manuscript to maintain the objectivity and accuracy of the Results presentation.

Comment 12: In the Discussion, the statement “However, no study has explored the relationship between AHEI and kidney stones” may be misleading. To my knowledge, prior studies have investigated the association between AHEI and kidney stones. Please review the literature carefully.

Response: We thank the reviewer for this insightful comment. Upon re-examining the literature, we found that a recent study has indeed investigated the association between the Alternate Healthy Eating Index (AHEI) and kidney stone risk. This reference was published after our initial analysis and drafting of the manuscript, which led to its unintentional omission. We have now revised the statement to accurately reflect the existing evidence and have cited the relevant study in the Discussion section.

Comment 13: Please discuss the strengths of this study and its public health significance in the Discussion.

Response: Thank you very much for this valuable suggestion. We only discussed the shortcomings and did not conduct a perfect discussion on the advantages. The shortcomings of the discussion were followed by highlighting the public health significance of this study.

“The strengths of this study include the use of a large, nationally representative sample with standardized dietary and biochemical assessments, the simultaneous evaluation of four dietary indices, and the application of advanced mixture models (WQS and BKMR). From a public health perspective, our findings emphasize that promoting healthy eating patterns could reduce kidney stone risk partly through uric acid regulation, supporting dietary modification as a cost-effective prevention strategy.”

Comment 14: The figure legends lack essential information, such as covariates adjusted in the models and explanations of abbreviations in the figures. This may affect reader comprehension. Please provide the missing details.

Response: We thank the reviewer for this helpful suggestion. We have revised all figure and table legends to make them more self-explanatory. Specifically, the legends now include details on the covariates adjusted in the regression models and full explanations of all abbreviations used. These modifications improve clarity and reader comprehension.

Formatting-related Issues

Comment 15: The use of spacing throughout the manuscript is inconsistent. Please check carefully to maintain uniform formatting.

Response: We appreciate this comment. The manuscript has been reformatted throughout using 12 font size and 22-point line spacing to ensure consistency and readability in accordance with journal formatting requirements.

Comment 16: The references contain multiple inconsistencies in title capitalization and journal abbreviations. Please carefully review and correct these issues.

Response: Thank you for pointing this out. We have thoroughly reviewed and standardized all references according to the journal’s formatting guidelines, ensuring consistency in title capitalization, punctuation, and journal name abbreviations.

Reviewer #2:

Comment 1: Did the authors check the normality of the data?

Response: Thank you for raising this important point regarding data normality. In this study, we employed an initial data quality check by examining the maximum and minimum values for all continuous variables to identify any potential outliers or data entry errors. No anomalous values were found that fell outside biologically or clinically plausible ranges, which we attribute to the rigorous data management and quality control procedures inherent to the NHANES database.

Comment 2: Can dietary apps based on artificial intelligence help reduce the risk of kidney stones? Adding this discussion point can be helpful in managing individual diets.

Response: We sincerely thank the reviewer for this insightful suggestion. We agree that artificial intelligence (AI)-based dietary applications hold great potential for improving personalized nutrition management. Such tools have been successfully applied in weight control, dietary assessment, and chronic disease prevention. By integrating AI algorithms to monitor total food intake, estimate nutrient composition, and evaluate dietary indices over time, these applications could help individuals maintain healthier eating patterns and potentially reduce the risk of kidney stones. Therefore, in the revised "Discussion" section, we added new content to the discussion, exploring the potential role that AI-driven diet apps can play in individualized prevention and management of kidney stones.

Reviewer #3: The study is actually interecting but it is needing some revision as:

Comment 1: please explain the reason authors needs to divide the model into crude, 1 and 2.

Response: Thank you for your insightful comment. We divided the logistic regression analysis into three models to assess the stability and independence of the associations between dietary indices and kidney stones. Specifically, the Crude model represents the unadjusted association; Model 1 adjusts for basic demographic factors (age, sex, and race); and Model 2 further adjusts for socioeconomic and clinical covariates (education level, marital status, BMI, cotinine, alcohol consumption, hypertension, and diabetes). This stepwise adjustment approach allows us to examine whether the observed associations persist after controlling for potential confounding factors and to better understand how each set of variables influences the relationship between dietary indices and kidney stone risk. We have clarified this explanation in the Methods section.

Comment 2: in the table 1 It would be more presentable if the author put age range.

Response: Thank you very much for this question. The age range (20–80 years) has now been clearly indicated in Table 1.

Comment 3: in the table 1 it would be more presentable if the author put BMI range.

Response: Thank you very much for this question. The BMI ranges have been added as follows: overall and non-stone group (13.18–84.40), and stone group (15.10–67.83). These values are now included in the revised Table 1.

Comment 4: Please provides high-quality in every figure.

Response: Thank you for your comments on the Figures. We have carefully checked the image quality, and there should be no ambiguity. The pixels of the 5 Figures in this study are all 300bpi, and

---

## [Decision Letter · Decision Letter 1]

18 Nov 2025

PONE-D-25-41276R1Uric acid levels mediate the association between four dietary indices and kidney stones in US adults: a cross-sectional study of NHANES 2007-2018PLOS ONE

Dear Dr. cao,

Thank you for submitting your manuscript to PLOS ONE. After careful consideration, we feel that it has merit but does not fully meet PLOS ONE’s publication criteria as it currently stands. Therefore, we invite you to submit a revised version of the manuscript that addresses the points raised during the review process.

Dear Authors, the reviewers have again suggested some major revisions. I recommend you revise the manuscript carefully based on the comments for reconsideration by the referees and subsequent decision.

We look forward to receiving your revised manuscript.

Kind regards,

Yogesh Kumar Jain, PhD

Academic Editor

PLOS ONE

Journal Requirements:

Reviewers' comments:

Reviewer's Responses to Questions

**Comments to the Author**

1. If the authors have adequately addressed your comments raised in a previous round of review and you feel that this manuscript is now acceptable for publication, you may indicate that here to bypass the “Comments to the Author” section, enter your conflict of interest statement in the “Confidential to Editor” section, and submit your "Accept" recommendation.

Reviewer #2: All comments have been addressed

Reviewer #3: All comments have been addressed

2. Is the manuscript technically sound, and do the data support the conclusions?

Reviewer #2: Yes

Reviewer #3: Yes

3. Has the statistical analysis been performed appropriately and rigorously? 

Reviewer #2: Yes

Reviewer #3: No

4. Have the authors made all data underlying the findings in their manuscript fully available?

Reviewer #2: Yes

Reviewer #3: Yes

5. Is the manuscript presented in an intelligible fashion and written in standard English?

Reviewer #2: Yes

Reviewer #3: No

6. Review Comments to the Author

Reviewer #2: (No Response)

Reviewer #3: Please provide different age groups and BMI group range (ex: age: 20-30, 30-40, 40-50,50-60,60-70 or < 50 and > 50 (and reason behind that) BMI: < 18.5, 18.5 - 24.9, 25 - 29.9, 30-39.9, and > 40), instead of only stating the range and mean number. Providing age groups (not only stating the overall age range), might help enhance the discussion since it affects the BMI and might affect the overall result of this study. Thus, necessary to be stated and discussed.

7. PLOS authors have the option to publish the peer review history of their article (what does this mean? ). If published, this will include your full peer review and any attached files.

**Do you want your identity to be public for this peer review?** For information about this choice, including consent withdrawal, please see our Privacy Policy .

Reviewer #2: No

Reviewer #3: **Yes: ** Besut Daryanto

---

## [Author Response · Author response to Decision Letter 2]

24 Nov 2025

Dear Editors and Reviewers:

Thank you for your letter and for the reviewers’ comments concerning our manuscript entitled “Uric acid levels mediate the association between four dietary indices and kidney stones in US adults: a cross-sectional study of NHANES 2007-2018” (ID: PONE-D-25-41276). Those comments are all valuable and very helpful for revising and improving our manuscript, as well as the important guiding significance to our researches. We have studied comments carefully and have made correction which we hope meet with approval. Revised portion are marked in red in the paper. The main corrections in the paper and the responds to the reviewer’s comments are as flowing:

Responds to the reviewer’s comments:

Reviewer #3:

Comment 1: Please provide different age groups and BMI group range (ex: age: 20-30, 30-40, 40-50,50-60,60-70 or < 50 and > 50 (and reason behind that) BMI: < 18.5, 18.5 - 24.9, 25 - 29.9, 30-39.9, and > 40), instead of only stating the range and mean number. Providing age groups (not only stating the overall age range), might help enhance the discussion since it affects the BMI and might affect the overall result of this study. Thus, necessary to be stated and discussed.

Response: Thank you very much for your valuable comment. We apologize for the misunderstanding in our previous revision. In the updated manuscript, we have revised the demographic data presentation according to your suggestion. Specifically, age has now been categorized into two groups (<60 years and ≥60 years) based on widely accepted WHO criteria for defining older adults, which may better reflect differences related to the elderly status. In addition, BMI has been regrouped following WHO standards into <18.5 (underweight), 18.5–25 (normal weight), 25–30 (overweight), and ≥30 (obesity). These subgroup classifications have been incorporated into both the Results and Discussion sections to better explore their potential influence on BMI-related outcomes and overall study findings.

In addition, the editor's summary of the reviewers' opinions and the overall evaluation of the article are also very helpful, and we have followed the suggestions to revise it one by one.

In all, I found the reviewer’ s comments and the editor's summary are quite helpful, and I revised my paper point-by-point. Thank you and the review again for your help! We tried our best to improve the manuscript and made some changes in the manuscript. These changes will not influence the content and framework of the paper. And here we did not list the changes but marked in red in revised paper.

We appreciate for Editors/Reviewer’ warm work earnestly, and hope that the revision will meet with approval. Once again, thank you very much for your comments and suggestions.

---

## [Decision Letter · Decision Letter 2]

14 Dec 2025

Uric acid levels mediate the association between four dietary indices and kidney stones in US adults: a cross-sectional study of NHANES 2007-2018

PONE-D-25-41276R2

Dear Dr. cao,

We’re pleased to inform you that your manuscript has been judged scientifically suitable for publication and will be formally accepted for publication once it meets all outstanding technical requirements.

Kind regards,

Yogesh Kumar Jain, PhD

Academic Editor

PLOS One

Additional Editor Comments (optional):

Reviewers' comments:

Reviewer's Responses to Questions

**Comments to the Author**

1. If the authors have adequately addressed your comments raised in a previous round of review and you feel that this manuscript is now acceptable for publication, you may indicate that here to bypass the “Comments to the Author” section, enter your conflict of interest statement in the “Confidential to Editor” section, and submit your "Accept" recommendation.

Reviewer #2: All comments have been addressed

Reviewer #3: All comments have been addressed

2. Is the manuscript technically sound, and do the data support the conclusions?

Reviewer #2: Yes

Reviewer #3: Yes

3. Has the statistical analysis been performed appropriately and rigorously? 

Reviewer #2: Yes

Reviewer #3: Yes

4. Have the authors made all data underlying the findings in their manuscript fully available?

Reviewer #2: Yes

Reviewer #3: Yes

5. Is the manuscript presented in an intelligible fashion and written in standard English?

Reviewer #2: Yes

Reviewer #3: Yes

6. Review Comments to the Author

Reviewer #2: (No Response)

Reviewer #3: Dear author,

Actually overall the manuscript revised is good, but the author has not provided the high-quality figure yet.

Best regards,

7. PLOS authors have the option to publish the peer review history of their article (what does this mean? ). If published, this will include your full peer review and any attached files.

**Do you want your identity to be public for this peer review?** For information about this choice, including consent withdrawal, please see our Privacy Policy .

Reviewer #2: No

Reviewer #3: **Yes: ** Dr. dr. Besut Daryanto, Sp.B., Sp.U(K)

---

## [Editor Report · Acceptance letter]

PONE-D-25-41276R2

PLOS One

Dear Dr. Cao,

I'm pleased to inform you that your manuscript has been deemed suitable for publication in PLOS One. Congratulations! Your manuscript is now being handed over to our production team.

Kind regards,

on behalf of

Dr. Yogesh Kumar Jain

Academic Editor

PLOS One